# Ability to Detect Digital Risks: Effects of an Educational Intervention and Dementia Risk Level

**DOI:** 10.3390/ijerph23010058

**Published:** 2025-12-31

**Authors:** Ricardo de Oliveira Ferreira, Isabella Gomes de Oliveira Karnikowski, Emmanuely Nunes Costa, Aline Gomes de Oliveira, Mariana Sodário Cruz, Iolanda Bezerra dos Santos Brandão, Margô Gomes de Oliveira Karnikowski

**Affiliations:** 1PPGCTS: Graduate Program in Health Sciences and Technologies, Faculty of Health Sciences and Technologies, University of Brasília–Ceilândia Campus, Brasília 72.220-260, DF, Brazil; margounb@gmail.com; 2Department of Economy, University of Porto, Porto 4200-375, Portugal; 3Program of Post-Graduation in Pharmaceutical Assistency, University of Brasília, Brasília 72.220-260, DF, Brazil; emmanuellynunes@gmail.com; 4School of Veterinary Medicine and Animal Science (FMVZ), Federal University of Uberlândia (UFU), Uberlândia 38410-337, MG, Brazil; 5Faculty of Health Sciences and Technologies, University of Brasília–Ceilândia Campus, Brasília 72.220-260, DF, Brazil; 6Euro-American University Center (Unieuro), Brasília 70200-001, DF, Brazil

**Keywords:** older adults, Addenbrooke’s Cognitive Examination, digital intervention

## Abstract

Introduction: Several studies have been conducted in the field of education for older adults, with an emphasis on teaching and learning processes related to the use of digital technologies. Among the relevant aspects to be considered in this context is the cognitive vulnerability of this age group in terms of digital security. Objective: The aim of this study was to analyze the relationship between cognitive aspects of older adults and their ability to identify digital risks, before and after participating in an educational intervention, as well as the effect of the intervention on cognition in this age group. Methodology: Analyses were conducted according to the educational intervention and control groups, further stratified by digital risk (SJT) and dementia risk, according to the ACE-R test. The Mann–Whitney test was used to identify possible differences in the likelihood of falling for digital scams, considering the dimensions generated by the simulations (SJT). Results: Overall, the educational intervention was effective for the media education dimension (delta −0.5), regardless of dementia risk. More specifically, a particular effect was observed in the post-intervention stage. Conclusions: The educational intervention was able to promote cognitive gains and reduce digital risks among older adults, particularly in the identification of misinformation, underscoring the importance of continuous and adapted programs to promote digital security in this population.

## 1. Introduction

The advancement of digital transformation has expanded access for different social groups, including older adults, to services, information, and technology-mediated activities. However, the growing presence of older individuals in digital environments takes place within a context marked by structural challenges, such as unequal access, digital exclusion, varying levels of technological literacy, and social dynamics that affect all age groups [1]. While the risk of online fraud victimization can occur at any age, it acquires specific features in older populations due to a combination of factors that include different levels of technological familiarity, previous experiences, access to social support, and, in some cases, changes in cognitive domains such as attention and executive functions [2]. It is therefore not an inevitable consequence of aging but rather a multifaceted phenomenon in which contextual conditions and the architecture of digital platforms play a significant role [3].

Several studies show that strategies used in digital scams often exploit cognitive and emotional vulnerabilities, but also factors such as social isolation, lack of protective policies, and barriers to reliable access [4,5].

For older adults, these issues may be intensified in contexts of low or no digital literacy or mobility restrictions, but they equally affect people with different profiles, highlighting the need for comprehensive education [6,7].

In response to this context, research highlights the importance of educational interventions and digital literacy programs aimed at preventing online fraud. Diverse strategies such as games, practical workshops, and personalized content have shown positive results both in risk recognition and in strengthening critical thinking and digital autonomy [8,9]. In addition, technology-mediated interventions have promoted greater social engagement, self-confidence, and maintenance of cognitive functions, which helps to address issues of isolation and disconnection of older adults from society [10].

The effects of these actions are not limited to technical or cognitive gains but extend to the promotion of self-esteem, a sense of belonging, and the reduction in social isolation. On the other hand, exposure to digital environments may generate new anxieties or conflicts related to security and risk management, which underscores the need for strategies that take into account the lived experiences of the target population [11,12,13,14,15].

In this regard, the research question guiding this study is as follows: ca personalized educational intervention improve older adults’ ability to identify digital risks, considering their specific cognitive characteristics? The hypothesis is that such an intervention will promote significant improvements in this identification, contributing to the strengthening of digital citizenship.

This study offers an original contribution by empirically analyzing the impact of an educational intervention focused on preventing digital fraud among older adults, considering specific cognitive variables such as attention and executive functions. By integrating cognitive aspects with personalized educational strategies, the research advances the existing literature by proposing a model that combines risk prevention, digital literacy, and the promotion of autonomy in a social group often neglected in public policies and technological projects.

For all the reasons stated, the objective of this study is to analyze the relationship between cognitive aspects of older adults and their ability to identify digital risks before and after participating in an educational intervention. The purpose is to contribute to the understanding of risk and safety factors in this context, as well as to reflect on and provide alternatives for the most effective strategies of prevention, support, and the promotion of digital citizenship for older adults.

## 2. Materials and Methods

The present study is a clinical trial with a follow-up in which the intervention group participated in an active learning methodology focused on digital literacy and data protection while the control group was assessed at the same points during the intervention period. The study was conducted in accordance with the Declaration of Helsinki and approved by the Research Ethics Committee on Human Subjects of the University of Brasília (CEP/UnB), opinion no. 7.326.551 [CAAE: 78064124.9.0000.8093], approved on [1 August 2025]. The experimental design is presented as shown in Figure 1. The sample was selected by convenience, and therefore using a non-probabilistic sampling [10], estimating the sample mean and standard deviation from the sample size, median, range, and/or interquartile range; see BMC Med Res Methodol. Dec 19, 2014;14(1):135 [11]).

Step 1. The older adults who participated in the present study were those initially selected by convenience and who then joined the course “Digital Literacy for Combating Property and Financial Violence and Media Education: Viva Mais Digital Citizenship Project.” Participation in the research occurred at the time of enrollment in the course, upon providing free and informed consent. The study included 101 individuals aged 60 years or older. This training initiative was organized by the National Secretariat for the Rights of Older Persons, under the Ministry of Human Rights and Citizenship, in cooperation with the University of Brasília (UnB). Its aim was to strengthen critical digital skills among older adults and promote citizenship within the context of information technologies in order to reduce digital risks.

Step 2. Anamnesis. For the sociodemographic characterization of participants, a structured questionnaire was used, adapted from the categories of the Brazilian Institute of Geography and Statistics (IBGE), and administered during the project enrollment period. Data collection was conducted individually, in a private setting, to preserve participants’ confidentiality and ensure appropriate conditions for administration. Researchers verbally applied the questionnaire and other instruments, recording responses manually, which ensured standardization of the procedure and facilitated participants’ understanding of the items. The sociodemographic questionnaire included variables such as sex, age, ethnicity, education, occupation, monthly income, and technology use, with particular emphasis on the use of mobile phones and computers.

Step 3. To assess digital risk, the Situational Judgment Test (SJT) was applied, as described by Lievens and Peeters and revised [16], administered at both pre- and post-intervention stages. Twelve stratified situations were used, distributed across four domains of digital risk.

To assess dementia risk, the Addenbrooke’s Cognitive Examination–Revised (ACE-R) [17] was used, adopting the cutoff score of 78 points as adapted [18], classifying scores < 78 as at risk and scores ≥ 78 as not at risk. The instrument was administered by researchers with training in psychology at both time points. The sample was subdivided into a control group with dementia risk (A1), control group without dementia risk (A2), educational intervention group with dementia risk (A3), and educational intervention group without dementia risk (A4) (Table 1). The twelve SJT situations used in the pre- and post-intervention assessments are listed in Table 2.

The remote activities applied only to the intervention group were monitored through attendance lists and online activity logs with course facilitators, with date and time records, as the course required a minimum participation of 75% of activities. The control group did not receive remote tasks.

### 2.1. Educational Intervention

This intervention consisted of a course entitled “Digital Literacy for Combating Property and Financial Violence and Media Education: Viva Mais Digital Citizenship Project.” It was designed based on the European DigComp framework [19,20] and grounded in participatory pedagogical practices. The course had a total duration of 32 h, distributed across eight in-person sessions of two hours each, interspersed with remote activities.

The program was divided into four thematic modules: property and financial violence, digital self-care, digital competence, and media education. The course content covered topics such as the definition and examples of property-related violence, legal foundations of information security (including the General Data Protection Law), financial education for online use, safe use of banking applications, basic computing and digital security, recognition and prevention of online scams, media education, and simulations of digital risk situations. The entire course was accompanied by mentors who supported older adults in carrying out the remote tasks.

### 2.2. Statistical Analysis

Statistical analyses were conducted using the SPSS 22 package for Windows (IBM Corp., Armonk, NY, USA). Numerical data related to age range were presented as means and standard deviations, while dementia risk scores and learning performance were presented as medians and interquartile ranges. Categorical data were presented as relative and/or absolute frequencies.

Comparisons of digital fraud risk scores, mediated by the independent variables dementia risk and control/intervention groups, were performed using the Mann–Whitney test. Differences in the frequencies of individuals at risk of dementia across different time points were assessed using McNemar’s test, while differences between the intervention and control groups were analyzed with the χ^2^ test for independent samples (likelihood ratio). The odds ratio (OR) for dementia risk classification was also calculated to verify the likelihood of dementia risk manifestation between intervention and control groups.

The study adopted an alpha level of 5% for all tests, except for the independent χ^2^ analysis, for which the confidence interval (CI) of −1.96 to +1.96 was used to determine statistical significance.

## 3. Results

### Participants

The sociodemographic characterization of the study sample indicated that the 101 older adult participants had a mean age of 68.7 ± 5.1 years, ranging from 60 to 81 years. The sample was predominantly composed of female participants (82.2%). Regarding self-declared race/ethnicity, the highest frequency was among those who identified as mixed-race (51.5%), followed by White (32.7%) and Black (13.9%) participants, while Asian represented only 1.0%, and 2.0% of participants chose not to report their classification.

With respect to educational level, the profile of participants was marked by the prevalence of individuals with completed high school (42.6%) or higher education (33.7%), indicating a sample with relatively high levels of schooling. Regarding occupational status, most participants were retired (64.4%), and a small portion (15.8%) reported being engaged in paid work at the time of data collection. The average monthly income reported by participants was BRL 4994.52 ± 4981.23, with a median of BRL 3000.00. Concerning technology use, 98.0% of participants regularly used a mobile phone, while 60.4% owned a computer. The results showed that the educational intervention produced statistically significant changes both in cognitive aspects and in the ability to identify digital risks among the older adult population studied.

The assessment of cognitive aspects using the ACE-R instrument revealed significant improvements in the intervention group, specifically in the domains of memory (*p* < 0.001), verbal fluency (*p* = 0.032), and language (*p* = 0.01), as presented in Table 3. No statistically significant differences were observed in the domains of attention and orientation (*p* = 0.736) or visuospatial abilities (*p* = 0.195).

The intervention also had a positive influence on participants’ dementia risk. The analysis revealed that, at the pre-intervention stage, the intervention group presented an odds ratio (OR) of 1.81 for dementia risk, which decreased to 0.5 after the period of educational intervention. This difference between the intervention and control groups was significant only at the post-intervention stage (*p* = 0.001), highlighting the effectiveness of the intervention in reducing this risk, as illustrated in Figure 2.

Regarding the ability of older adults to identify digital risks (Table 4), a significant reduction was observed only in the Media Education dimension (*p* = 0.019) after the educational intervention in the intervention group compared to the control group. The other evaluated dimensions (Property and Financial Violence, Digital Security, and Digital Competence) did not show statistically significant differences between the groups.

Analyzing the average digital risk, stratified by the presence or absence of dementia risk (Figure 3), it was found that participants at risk of dementia showed higher mean scores of digital risk when compared to those without risk, regardless of group or assessment time point. After the educational intervention, a trend toward a reduction in average digital risk was observed, especially in the intervention group with dementia risk, indicating a favorable effect of the intervention in mitigating these risks, as visually demonstrated in Figure 3.

Further exploring the relationship between digital risk and dementia (Table 5), it was observed that older adults at risk of dementia consistently presented higher scores in the dimensions of digital risk compared to those without risk in both groups (intervention and control), both before and after the intervention. Statistically significant differences were noted, as shown in Table 5, particularly in the domain of Property and Financial Violence (D1) pre- and post-intervention for the intervention group compared to the control group. Older adults at risk of dementia who participated in the educational intervention showed a significant improvement in the Media Education dimension (D4) after the intervention.

## 4. Discussion

The present study demonstrated that the intervention “Viva Mais Cidadania”, a short-term course, achieved the dual purpose of enhancing cognitive domains and reducing digital vulnerabilities among older adults. It was observed that half of the participants were at risk for dementia at the pre-test stage, contradicting the cognitive reserve theory, which posits that higher education and income are associated with lower cognitive decline [8]. This discrepancy suggests the influence of psychosocial factors, such as the pronounced social isolation following the pandemic, which may offset traditionally described protective effects [21].

From a population perspective, there is recent evidence that education reduces the risk of dementia over the life course. The Lancet Commission identifies low education as a modifiable risk factor, meaning that increasing years of schooling can help to prevent or delay some cases. Furthermore, an analysis of the UK Biobank indicates that continuous participation in adult education is associated with a lower incidence of dementia and greater hippocampal volume, pointing to a plausible mechanism for dementia prevention.

After eight in-person sessions interspersed with remote tasks, significant increases were observed in memory, verbal fluency, and language (Table 3), along with a reduction in the odds ratio for dementia risk from 1.81 to 0.50. These gains exceed the average reported for short-term interventions in recent reviews, where improvements of two to three points on the ACE-R are already considered clinically relevant [9,10]. Activities based on solving digital problems, small-group discussions, and individualized mentoring appear to have supported divided attention and information updating processes—mechanisms closely linked to the consolidation of episodic memory [4,18].

In line with the findings of the study conducted in Brazil, it was observed that a randomized clinical trial in the United Kingdom also showed that a ten-week course on learning to use tablets and applications resulted in significant improvement in processing speed among older adults, indicating that digital literacy interventions can produce measurable cognitive gains in the short term [22]. In the United States, similar results were observed with an iPad training program for older adults, which also led to improvements in cognitive performance among participants in the intervention group [23].

From a population perspective, there is recent evidence that education reduces the risk of dementia throughout life. The Lancet Commission identifies low education as a modifiable risk factor, meaning that increasing years of schooling may help to prevent or delay some cases. Furthermore, an analysis from the UK Biobank indicates that continuous participation in adult education is associated with a lower incidence of dementia and greater hippocampal volume, suggesting a plausible mechanism for dementia prevention. In addition, educational programs tend to expand social networks, sense of purpose, and social participation, which are factors consistently associated with lower dementia risk [22]. This pattern was also observed in the Viva Mais Digital Citizenship course.

However, our study found that half of the participants were at risk for dementia at the pre-test stage and that most had high levels of education and an average income at least three times higher than the Brazilian minimum wage, contradicting the cognitive reserve theory, which posits that higher education and income are associated with lower cognitive decline [8,9]. This discrepancy suggests the influence of psychosocial factors, such as the heightened social isolation in the post-pandemic period, which may neutralize the protective effects traditionally described [16].

With respect to digital risks, only the Media Education dimension showed a statistically significant reduction compared to the control group (Table 4). The literature indicates that content aimed at fact-checking and identifying fake news generates greater engagement, as it connects directly to everyday situations on social media [21]. This effect reinforces the argument that intrinsic motivation toward immediate topics facilitates the internalization of heuristics for source verification.

No changes were observed in the dimensions of Property and Financial Violence, Digital Security, and Digital Competence. Qualitative studies show that cybersecurity topics with technical terminology, such as multifactor authentication or app permissions, can provoke anxiety and feelings of inadequacy in older adults, making it harder to transfer learning [6]. Moreover, the increasing sophistication of fraud tactics amplifies vulnerability in this age group [21], which may explain the stability of scores even with the same exposure time.

Stratified analysis revealed distinct patterns depending on dementia risk (Table 1). Participants without risk showed greater reductions in Media Education scores, whereas those at risk showed only a modest improvement in Property and Financial Violence. Working memory deficits limit the generalization of complex knowledge in individuals with mild cognitive impairment [21,23], which is a phenomenon that helps to explain the lower gains in this subgroup.

The findings suggest that media literacy may serve as an entry point for broader digital protection programs. Improvement in misinformation detection is linked to civic participation and public health, as fraudulent campaigns frequently exploit financial and medical topics [7,15]. Thus, the benefits go beyond the individual sphere, with potential impacts on reducing economic losses and health risks.

In recent years, several studies have sought to develop educational strategies aimed at protecting older adults from digital and financial fraud. In Brazil, this investigation is grounded in an intervention proposal that combines digital citizenship and fraud prevention, with the goal of expanding risk recognition capacity and strengthening older adults’ confidence in the online environment. In line with this perspective, Chung and Yeung [9], in a study conducted in Hong Kong, evaluated the use of a board game as an educational resource and found significant benefits: participants reported greater awareness of fraud situations, increased self-efficacy, and reduced susceptibility to scams, including in a behavioral test simulating a phone call. The comparison between these two contexts points to an international trend toward using educational and playful methodologies as effective prevention strategies, reinforcing the relevance of the present study.

The persistence of deficits in the digital security and digital competence dimensions suggests the need for methodological adjustments that make these contents more easily assimilated. Evidence from hybrid studies indicates that breaking down technical topics into gradual micro-tasks, combined with accessible terminology and gamification mechanisms, increases knowledge retention among older adults by approximately 25% [23]. Furthermore, the literature shows that the everyday participation of family members or caregivers acts as a mediator of learning, facilitating the translation of abstract guidance into effective online protection practices [24]. Thus, it is clear that the program did not reach its full potential, and greater exposure frequency is needed to expand the impact precisely on the areas that proved more resistant to change [24].

The cognitive gains observed, when projected to population-level scenarios, have relevant implications for long-term care planning. Modeling based on international cohorts suggests that each additional point on the ACE-R can delay the need for specialized dementia-related care by three months on average [22]. When simultaneously considering the mitigation of financial losses resulting from digital fraud, an aggregate potential for cost savings emerges for health and social security systems, giving digital literacy interventions a strategic role in preventing future expenses.

Some methodological constraints, however, limit the interpretive scope of the results. The follow-up period was limited to three months, and exposure to digital scams was based on self-reports, which are subject to memory and social desirability biases. It is recommended that subsequent investigations adopt longitudinal designs with at least one year of follow-up and include objective indicators, such as banking records of phishing attempts or official fraud alerts, as noted [10]. In addition, comparisons between in-person, remote, and hybrid formats may clarify the cost-effectiveness relationship in different socioeconomic contexts.

The high initial prevalence of dementia risk, even among participants with higher education and income, highlights the multifactorial complexity of cognitive decline and underscores the importance of prior screening in the design of interventions. Adjusting the depth and pace of activities to the cognitive profile of each group may enhance both learning and the maintenance of safe practices. In this sense, short but flexible programs appear to be a viable alternative to achieving broad population coverage without sacrificing pedagogical effectiveness.

## 5. Limitations and Future Directions

This study used a convenience sample composed of voluntary participants with relatively high education levels, which reduces representativeness and limits generalizability to other contexts. No a priori sample size calculation was performed. Follow-up was short, with no reassessment at three or six months, as the course lasted only one month. For future studies, probabilistic sampling and greater sociocultural diversity are recommended, along with prior sample size calculation, follow-up periods of 6 to 12 months, inclusion of objective outcomes (for example, banking records of fraud attempts and digital security indicators), and testing of different “dosages” and reinforcement strategies for the intervention.

## 6. Conclusions

The digital literacy intervention demonstrated potential to integrate cognitive promotion and online safety within the same educational protocol by reducing dementia risk and improving the ability to discern questionable media content. These results support the assumption that structured digital activities, even in short-term programs, can contribute to delaying cognitive decline while simultaneously fostering protective attitudes against online fraud.

The high prevalence of ACE-R scores below the cutoff point, even among participants with higher education and income, highlights that prior cognitive screening should be a mandatory step in the planning of interventions for older adults. Such screening makes it possible to adjust the pace, depth, and format of content to the typical limitations of attention, working memory, and processing speed associated with aging, thereby increasing the likelihood of adherence and sustained gains. Thus, future programs should combine initial neuropsychological assessment, gradual pedagogical strategies, and support from family members or caregivers to maximize educational effectiveness and protect older adults in an increasingly complex digital environment.

Although contextualized methodologies favor engagement, cybersecurity topics of greater technical complexity proved less sensitive to change. The persistence of these deficits suggests that instructional strategies focused on conceptual simplification, incremental gamification, and family or caregiver support should be incorporated into future programs. Moreover, the predominance of self-reported measures and the short follow-up period limit the strength of inferences regarding the permanence of gains, pointing to the need for the use of objective indicators (for example, banking records of phishing attempts) and longitudinal designs with multiple waves of assessment.

Future research should include heterogeneous cohorts in terms of education, income, physical health, and social support, as well as compare in-person, remote, and hybrid formats from a cost-effectiveness perspective. By integrating successive neuropsychological assessments and robust metrics of real exposure to scams, such studies will be able to clarify the durability of the observed effects and identify critical moderators of effectiveness. These findings are essential to guide public policies aimed at promoting active aging that is both cognitively protected and digitally safe.

## Figures and Tables

**Figure 1 ijerph-23-00058-f001:**
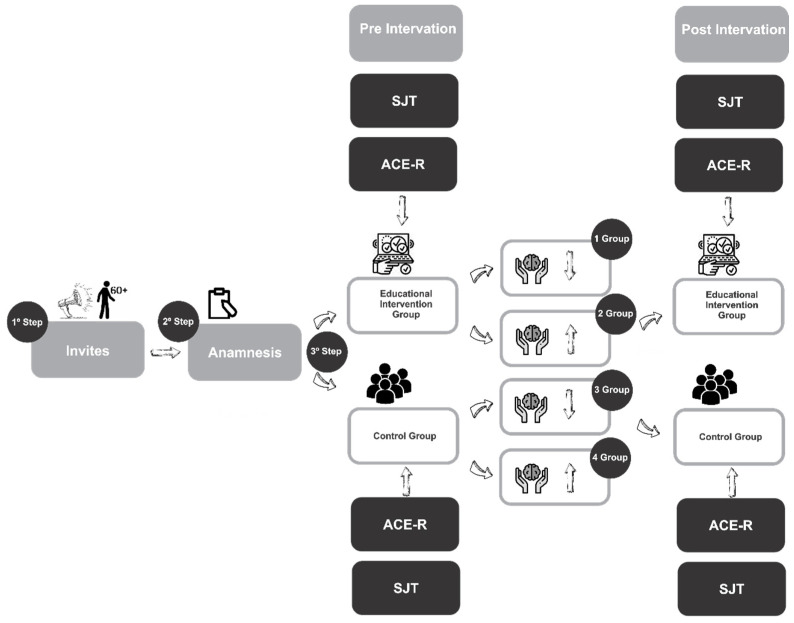
Experimental design of the study.

**Figure 2 ijerph-23-00058-f002:**
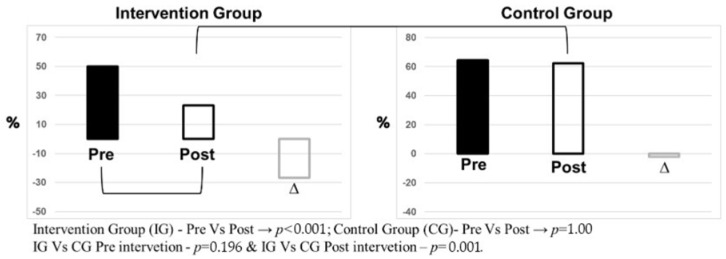
Comparison of the relative frequency of dementia risk and its kinetics in the intervention group (*n* = 56) and control group (*n* = 45), before and after the intervention.

**Figure 3 ijerph-23-00058-f003:**
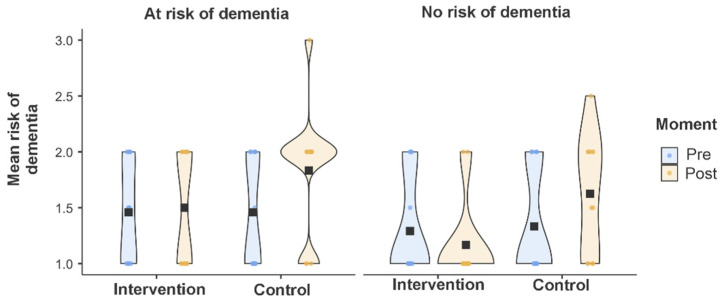
Average digital risk, stratified by dementia risk in the pre- and post-intervention periods.

**Table 1 ijerph-23-00058-t001:** Stratification of participants according to dementia risk (ACE-R cutoff: <78) and participation in the educational program.

Group	Study Condition	Dementia Risk Status (ACE-R)	*n*
A1	Control	At risk (<78)	16
A2	Control	No risk (≥78)	29
A3	Intervention	At risk (<78)	28
A4	Intervention	No risk (≥78)	28

**Table 2 ijerph-23-00058-t002:** Stratified situations by dimensions for assessing the degree of digital risk in the pre- and post-educational intervention applied to individuals aged 60 years and older.

Dimension	Pre-Intervention Situations	Post-Intervention Situations
Financial and/or Property-Related Abuse	Situation 1. You access a popular shopping website that is offering impressive discounts on high-value items. However, the only available payment method is via bank transfer or Pix, with no option for refund or return. What would you do in this situation?	Situation 1. You receive a WhatsApp message from a close relative, but the message is sent from an unfamiliar number. The person asks urgently for money to pay a bill, claiming they are using a temporary phone because their main device was stolen. How would you respond?
	Situation 2. You receive a phone call from someone claiming to be a bank employee. They ask you to confirm your personal banking information, stating that it is necessary to avoid the imminent blocking of your account. How would you respond?	Situation 2. You receive the following email: “Hi! Find out how you can quadruple your investment in just one month! Click the link below to learn more about this innovative platform and start earning today! Don’t miss this chance!” www.superprofitableinvestment.com Click here. How would you react to this email?
	Situation 3. You receive a private message on Facebook from someone you don’t know. They introduce themselves as a representative of an organization that provides financial assistance to people in economic hardship. The message is well written and includes a professional-looking logo. To “activate” the assistance process, the person asks you to make a small bank transfer to cover what they describe as “necessary administrative fees.” How would you proceed?	Situation 3. You see an ad on Instagram offering a free online course with a certificate. When you click on the link, you are asked to provide personal information such as your ID number, tax ID, and credit card number to “validate” the certificate. How would you proceed?
Digital Security	Situation 4. You received an email saying that your email account is about to be deactivated. The email asks you to click on a link to verify your account. How would you proceed?	Situation 4. You are using a public computer to access your email. When finishing, you realize you forgot to log out. How would you proceed?
	Situation 5. You accessed a shopping website and noticed that the address starts with “http://”. How would you proceed?	Situation 5. When installing a new app on your phone, it asks for access to your camera, contacts, and location—even though these functions are not required to use the app. How would you proceed?
	Situation 6. You found a USB stick on the ground near your home and are curious about its contents. How would you proceed?	Situation 6. You realize that your social media account was accessed from an unknown location. How would you proceed?
Digital Competency	Situation 7. Mrs. Cida is organizing recipes she found on the internet. She saved them with generic names like “file1.pdf” and “recipe2.docx”, and now she’s having trouble locating the ones she wants to use. How can Dona Cida better organize her files?	Situation 7. You are looking for information on how to lower blood pressure. You find two articles: one on a well-known cardiology website with scientific references and identified authors, and another on a personal blog with vague comments and promises of a “miracle cure.” Which article seems more reliable to use?
	Situation 8. Mrs. Cida wants to access her banking app but realizes she uses the same password for various accounts, such as social media and email. She heard this could be risky. How can Dona Cida make her accounts more secure?	Situation 8. You joined a Facebook gardening group. During a conversation, some members made rude or disrespectful comments that disrupted the discussion. Question: How can you handle this situation to maintain good communication in the group?
	Situation 9. Mr. Antônio noticed that his computer freezes during a video call with his family. He wants to find a way to continue the conversation. What can Seu Antônio do to fix the problem?	Situation 9. You are preparing a presentation for your community group. During your research, you find free images you’d like to use. Some of them say “free for personal use only,” and others mention licenses. Question: How can you make sure you are using the images legally?
Media Literacy	Situation 10. Mr. Antônio saw a graph in a newspaper article stating that energy consumption increased by 200%. The graph is not clear, making it hard to interpret the data. What can Seu Antônio do to correctly interpret the graph?	Situation 10. You watch a video on a social media platform showing a politician making a controversial statement that promotes racial and financial discrimination. How would you proceed?
	Situation 11. Mrs. Ana wants to participate in an online forum about retirement. During the conversation, some participants begin to spread false information about changes in the pension system. How can Dona Ana contribute to a more productive discussion?	Situation 11. You see a post on social media claiming that a new virus is being spread through food sold in supermarkets. The post has already been shared thousands of times. How would you proceed?
	Situation 12. Mr Paulo is part of a Facebook group on local history. He saw another participant sharing a photo without mentioning the source. What can Seu Paulo do to encourage best practices in the group?	Situation 12. You receive a WhatsApp message saying that drinking hot water with lemon every morning cures serious illnesses. The message claims the information came from a famous doctor. How would you proceed?

**Table 3 ijerph-23-00058-t003:** Medians and interquartile ranges (IQRs) of scores in the cognitive domains assessed by the ACE-R, before and after the educational intervention, in the intervention group.

Cognitive Domain	Pre-Intervention	Post-Intervention	Δ	*p* (Wilcoxon)
Attention and Orientation	16 [14;25;18]	17 [15;18]	0 [−1;1]	0.736
Memory	19 [16;21]	21 [18;24]	3 [0;5]	<0.001
Fluency	8.5 [7;11]	10 [8,25;11,75]	1 [−1;3]	0.032
Language	23 [21;25;25]	25 [23;26]	1 [−1;2]	0.01
Visuospatial	13 [11;15]	14 [12;16]	0.5 [1;2]	0.195
Total Score	79 [70.5;87.75]	85.5 [79;91.75]	4 [0.75;12.5]	<0.001

Includes Δ values (difference between pre- and post-intervention), Wilcoxon test, and Friedman test. Source: research data.

**Table 4 ijerph-23-00058-t004:** Presentation of the dimensions related to digital risk situations and the intervention and control groups.

Dimension	Control	Intervention	*p*
Property and/or Financial Violence	0.0 [−1;0.0]	−1 [−1;0.0]	0.292
2. Digital Security	0.0 [0.0;1]	0.0 [−1;1]	0.111
3. Digital Competence	1 [−0.5;1]	0 [−0.75;1]	0.659
4. Media Education	0.0 [−1;1]	−0.5 [−1;0.0]	0.019

**Table 5 ijerph-23-00058-t005:** Digital risks and dementia: comparative analysis between the intervention and control groups.

		D1	D2	D3	D4
Pre-Intervention					
No Dementia Risk	Intervention	2 [1;2.5] *	1 [1;2]	1 [0;2]	2 [1;2]
	Control	3 [2;3]	1 [1;2]	1.5 [0;2]	1 [0;2]
With Dementia Risk	Intervention	2 [2;3] *	1 [1;2]	1 [0;1]	1 [0;2]
	Control	3 [2;3]	1 [1;2]	1 [0;2]	1 [0;2]
Post-Intervention					
No Dementia Risk	Intervention	2 [1;2] *	1 [1;2]	2 [0;2]	1 [1;2]
	Control	3 [2;3]	2 [1;2]	2 [1;2]	1 [0;2]
With Dementia Risk	Intervention	2 [1;2] *	1 [1;2]	1 [1;2]	1 [0;1] *
	Control	2 [2;3]	2 [1;2]	1 [1;2]	1 [0;2]
Change (Δ)					
No Dementia Risk	Intervention	−1 [−1;0]	1 [1;2]	2 [0;2]	1 [1;2]
	Control	0 [−1;0]	2 [1;2]	2 [1;2]	1 [0;2]
With Dementia Risk	Intervention	/ [00]	1 [1;2]	1 [1;2]	1 [0;1]
	Control	/ [0;0]	2 [1;2]	1 [1;2]	1 [0;2]

Legend: values are presented as median [IQR] of the SJT. D1 = Media Education; D2 = Digital Security; D3 = Digital Competence; D4 = Property and Financial Violence. The asterisk (*) symbolizes *p* < 0.05 vs. Control at the same time point.

## Data Availability

The original contributions presented in this study are included in the article material. Further inquiries can be directed to the corresponding author(s).

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
