# Peer review of "Ability to Detect Digital Risks: Effects of an Educational Intervention and Dementia Risk Level"

_ijerph, 2025, doi:10.3390/ijerph23010058_

Round 1

Reviewer 1 Report

Comments and Suggestions for Authors

Thank you for the opportunity to review this manuscript, which explores the relationship between cognitive functioning and digital risk identification among older adults through an educational intervention.

The article addresses a relevant and timely issue, employs a validated assessment strategy, and benefits from a structured intervention based on the DigComp framework. The study design includes a control group, enhancing the credibility of the results. However, several aspects require clarification or improvement to strengthen the rigor and impact of the findings:

  • The ACE-R needs to be explained more clearly: a cut-off score of 78 is mentioned and the sample is said to be divided into four groups (lines 102–108), but these are not reflected in Table 1 as stated
  • How were remote activities monitored or controlled? (line 117)
  • It would be advisable to include a table presenting the sociodemographic results.
  • The discussion is underdeveloped. It should be enriched by incorporating similar international studies to support and contrast the findings.
  • The article does not report any sample size calculation. It also fails to justify whether the 101 participants were sufficient to detect significant differences between the variables analysed. This is a major methodological weakness, particularly in a study involving group comparisons.
  • The study’s limitations should be included: the sampling method and its representativeness (voluntary sample, high educational profile), the absence of a sample size calculation, the limited generalisability to other sociocultural contexts, and the lack of follow-up (e.g., at 3 or 6 months), which prevents assessment of medium- to long-term effects.
  • The name of the Research Ethics Committee that approved the study, the protocol code, and the date of approval should be specified (lines 74–75; 271–273).
  • Please revise the English in Figure 1 (“Pre intervation, Anamnese”).
  • Tables 2 and 4 are not referenced in the text.
  • Please revise the English in Table 4 (“Pré intervation”). What do the asterisks indicate?
  • Figure 3 is not in English and is not explained in the text.

I hope these comments are helpful as you revise your manuscript.

Yours sincerely,

Reviewer 2 Report

Comments and Suggestions for Authors

This study examined how cognitive abilities affect older adults’ ability to spot digital risks before and after a digital literacy training. Participants were grouped by dementia risk (using ACE-R) and tested on risk awareness (via SJT). After the training (focused on media literacy, security, and fraud prevention), all groups improved—especially in media literacy—showing that tailored education boosts both digital safety skills and cognitive engagement. The findings highlight the need to adapt digital literacy programs for different cognitive levels in aging populations.

The sample was overwhelmingly female (82.2%), which may limit the generalizability of findings to older adult populations with more balanced gender distribution.

A large proportion had completed high school (42.6%) or higher education (33.7%), while only 12.9% had elementary-level education. This suggests a possible bias toward more educated participants, which may not reflect the broader elderly population.

The reported mean monthly income (R$ 4,994.52) is notably high for many elderly populations, particularly in Russia, where pensions and retirement incomes are often lower. The high standard deviation (SD = R$ 4,981.23) indicates extreme income variability, raising concerns about economic representativeness.

While diverse, the small sample sizes for some groups (e.g., Asian: 1.0%) make subgroup analyses unreliable – Lines 145-146

Dementia risk reduction (OR = 0.5 post-intervention) is a strong finding. But there is a question on why did education lower dementia risk? (Neuroprotective? Social engagement?). Highly educated, higher-income sample may not reflect general elderly population. The study tackles a relevant, underexplored niche (digital literacy → dementia risk) but would benefit from deeper mechanistic insights, longer-term follow-up, more diverse sample representation. Overall, it provides promising preliminary evidence but requires further validation for broader applicability

Reviewer 3 Report

Comments and Suggestions for Authors

The article is based on the issue that ‘the growing integration of older adults into digital environments raises concerns about their ability to recognize and manage digital risks, particularly given the cognitive vulnerabilities common in this population.’ This main question highlights the timeliness and relevance of the research for the fields of knowledge involved, aligning with the journal’s scope and demonstrating its potential to engage readers’ interest.

Nevertheless, it is considered that the study’s original contributions could be more clearly emphasized, particularly to highlight its additions to the subject area in comparison with other published material. This would enhance the clarity regarding how it addresses gaps in the interpretation of the multifaceted phenomenon under analysis.

Some of these suggestions could, at the authors’ discretion, also be incorporated into the abstract, despite its currently adequate structure. The keywords are relatively diverse in relation to the title, which positively expands the indexing possibilities for the manuscript.

In Section 1 (Introduction), the contextualization of the problem and the presentation of justifications are clear, as is the formulation of the general objective, which is ‘to analyze the relationship between cognitive aspects of older adults and their ability to identify digital risks, before and after an educational intervention.’ However, this part of the text lacks a more explicit presentation of a guiding research question and/or a hypothesis to be tested. On the other hand, the references used in this part of the text are almost exclusively from the past five years (over 90%), which reflects a strong emphasis on recent literature.

Section 2 (Materials and Methods) presents aspects of the implementation of a controlled design, with participants (101 older adults, with a mean age of 68.7 years, ranging from 60 to 81) stratified by dementia risk based on the Addenbrooke’s Cognitive Examination (ACE-R) and digital risk identification assessed using the Situational Judgment Test (SJT). From the Figure 1, the investigative steps (invites, anamnesis, and pre- and post-intervention) are sufficiently detailed to ensure the study’s reproducibility, including elements of the educational intervention and statistical analysis. It is worth noting that the intervention consisted of a comprehensive digital literacy program focused on media literacy, digital security, and financial abuse prevention.

According to the authors, ‘the entire study followed Brazilian regulations on current ethical standards and was approved under ethics committee opinion number 7.326.551.’ In this section detailing methodological procedures, new references are introduced, approximately 75% of which are from the past five years.

Accompanied by figures and tables to ensure proper understanding of the content, the results (Section 3) present the characterization of participants with and without dementia risk, as well as of the intervention and control groups at both pre- and post-intervention stages. Although concise, the presentation of findings demonstrates relative degree of scientific rigor.

As stated by the authors, the ‘results demonstrated significant improvements in media literacy across all participants, regardless of dementia risk, with cognitive benefits evident post-intervention. These findings indicate that tailored educational strategies effectively enhance digital risk awareness and cognitive engagement among older adults.’

Section 4 (Discussion) deepens the analytical approaches to the results, drawing on current references cited in the introduction, along with additional sources. Consequently, the debate is grounded primarily in literature published within the last five years (approximately 85%).

The conclusions are consistent with the evidence and arguments presented. They effectively address the main research question. As the authors point out, ‘the study underscores the necessity of considering cognitive status when designing digital literacy programs to promote autonomy and reduce vulnerability. Further research is needed to explore long-term impacts and adapt interventions to diverse cognitive profiles within the aging population.’ However, the investigative limitations could be further elaborated, along with more detailed recommendations for future studies on the topic.

As previously noted, the references are appropriate. A noteworthy aspect of the manuscript lies in the recency of its sources, with nearly 95% published within the last decade and approximately 85% within the past five years.

Finally, the figures and tables are considered pertinent. Nonetheless, for Figure 1, it is suggested—though not mandatory—that a note be added to clarify the meaning of the acronyms STJ and ACE-R. Additionally, Figure 3 should be translated into the language of the manuscript.

Comments on the Quality of English Language

The manuscript’s writing is clear, but it requires some revision adjustments regarding spelling, grammar, and typing accuracy.

Round 2

Reviewer 1 Report

Comments and Suggestions for Authors

Dear Authors,

Thank you for addressing my previous comments and for the revisions made to the manuscript. After reviewing the updated version, I have only a couple of final remarks:

  1. In line 200, it states Table 1 when it should state Table 2.
  2. Please review all decimal numbers throughout the manuscript, including the tables, and replace them with the symbol "."

Sincerely,
